# Multiomic Investigations into Lung Health and Disease

**DOI:** 10.3390/microorganisms11082116

**Published:** 2023-08-19

**Authors:** Sarah E. Blutt, Cristian Coarfa, Josef Neu, Mohan Pammi

**Affiliations:** 1Department of Molecular Virology and Microbiology, Baylor College of Medicine, Houston, TX 77030, USA; sb691007@bcm.edu; 2Department of Molecular and Cellular Biology, Baylor College of Medicine, Houston, TX 77030, USA; coarfa@bcm.edu; 3Dan L Duncan Comprehensive Cancer Center, Baylor College of Medicine, Houston, TX 77030, USA; 4Department of Pediatrics, Section of Neonatology, University of Florida, Gainesville, FL 32611, USA; neuj@peds.ufl.edu; 5Department of Pediatrics, Section of Neonatology, Baylor College of Medicine and Texas Children’s Hospital, Houston, TX 77030, USA

**Keywords:** multiomics, lung, pulmonary, disease models, machine learning

## Abstract

Diseases of the lung account for more than 5 million deaths worldwide and are a healthcare burden. Improving clinical outcomes, including mortality and quality of life, involves a holistic understanding of the disease, which can be provided by the integration of lung multi-omics data. An enhanced understanding of comprehensive multiomic datasets provides opportunities to leverage those datasets to inform the treatment and prevention of lung diseases by classifying severity, prognostication, and discovery of biomarkers. The main objective of this review is to summarize the use of multiomics investigations in lung disease, including multiomics integration and the use of machine learning computational methods. This review also discusses lung disease models, including animal models, organoids, and single-cell lines, to study multiomics in lung health and disease. We provide examples of lung diseases where multi-omics investigations have provided deeper insight into etiopathogenesis and have resulted in improved preventative and therapeutic interventions.

## 1. Introduction

Respiratory diseases account for over 5 million deaths yearly, constitute a significant cause of morbidity and are a huge burden to healthcare systems worldwide [1,2]. Annually, about three million deaths are due to chronic obstructive pulmonary disease and asthma (the most common disease) accounts for half a million deaths. Other pulmonary diseases with chronic inflammation and obstruction and often exacerbated by infection include cystic fibrosis, idiopathic pulmonary fibrosis, ciliary dyskinesia, and pneumonia are also leading causes of death and lung cancer leads to the cancer-related deaths category [1,2].

With the ongoing global pandemic due to SARS-COVID-19 [3], respiratory diseases remain a leading cause of death and disability. Recent advances in high-throughput technologies have provided access to multiomics biological data, including genomics, epigenomics, transcriptomics, proteomics, metabolomics and immunomics, and provide a holistic view of pathophysiology in lung disease [4]. Multiomics data give a comprehensive overview of cellular processes (e.g., gene transcription, protein translation or epigenetic processes) associated with a disease and an insight into the complexity of the disease. Biological insights from multi-omics can be integrated with clinical and social data and applied in the clinical setting for improved health outcomes. Single omics are limited by providing associations, whereas multiomic integrations result in deriving a holistic insight that generates testable hypotheses about mechanisms.

State-of-the-art machine-learning methods can integrate high dimensional omics datasets resulting in the ability to predict short- and long-term health trajectories and enable early timely interventions that alter the health course towards better outcomes (precision medicine) [5]. Large datasets such as the omics dataset rely on ‘deep learning’ based on neural networks loosely modeled after neurons of the brain [6]. The insights gained by deep learning of multiomic datasets enhance personalized healthcare decision-making (precision medicine) and biomarker discovery [5,7].

The NIH defines precision medicine as ‘an innovative approach that takes into account individual differences in patients’ genes, environments, and lifestyles [8] (https://www.nih.gov/about-nih/what-we-do/nih-turning-discovery-into-health/promise-precision-medicine (accessed on 18 August 2023)). There is an urgent need to shift our current thinking on traditional reactive medicine based on prior literature/data to a more proactive precision medicine (PM) based approach where the trajectory towards health and disease can be predicted in advance, so interventions to improve survival or decrease morbidity can be instituted earlier to improve survival and decrease morbidity. Machine learning has already enabled a holistic systems biology approach in oncology for predicting survival, disease severity and biomarker development [9,10,11,12,13,14,15,16,17]. This proactive approach is being adapted in other fields and disciplines, including cardiovascular medicine and endocrinology [18,19,20]. In this review, we will assess the multi-omics strategy as it is integrated into human, animal and organoid models to provide insight into lung health and disease.

## 2. Insights into Cell Biology Using Multi-Omics

Quantitative omics technologies enable cost-effective and high-throughput profiling of various facets of cell biology (Table 1). Genomics can be profiled using whole-exome or whole-genome sequencing. Transcriptomics is assessed using RNA-Sequencing (RNA-Seq). Protein expression in its complete form or as post-translational modifications is measured using mass-spectrometry or antibody-based proteomics assays. Measurable epigenomic characteristics of the cells span DNA methylation (assessed via whole-genome bisulfite sequencing or probe-based micro-arrays), microRNAs (measured using smallRNA-Seq), histone modifications (measured using Chromatin Immunoprecipitation and sequencing (ChIP-Seq)), and open chromatin (measured using an assay for transposase-accessible chromatin with sequencing (ATAC-Seq)). Getting closer to cell biology, metabolomics and lipidomics can be measured using mass-spectrometry-based techniques. Appreciating that humans live in symbiosis with a rich microbial environment including bacterial, viral and fungal communities, the microbiome is evaluated by whole genome shotgun sequencing (WGS) or bacteria by 16S rDNA or fungi by 18S rDNA sequencing.

The advent of single-cell technologies, including single-cell RNA-Seq (scRNA-Seq) and single-cell ATAC-Seq (scATAC-Seq), have provided further insight into cell biology in the last decade. A single-cell multi-omics study on SARS-CoV2 employed single nuclei RNA-Seq and single nuclei ATAC-Seq in phenotypically healthy lungs of donors with ages of 30 weeks gestation, three years, and 30 years [21]. The main findings relate to identifying elements associated with transcriptional regulation, including SARS-CoV-2 host entry gene TMPRSS2 in alveolar type 2 cells, which had immune regulatory signatures and harbored variants associated with respiratory outcomes. The study aimed to decipher the cell-specific landscape of expression and candidate cis-regulatory elements (cCREs) of key genes for SARS-CoV2 infection associated with entry into the host cell, including ACE2 and TMPRSS2, and to further explore the changes with age, given the age-associated risks of SARS-CoV2 infection and outcome. ACE2 transcript was found in under 100 cells, almost half in AT2 cells. TMPRSS2 expression was detected particularly in AT1, AT2, club, ciliated, and goblet cells. ATAC signal, e.g. accessible chromatin, was found primarily in gene bodies for ACE2 and TMPRSS2 in AT1, AT2, club, ciliated, and basal cells. cCREs are areas of ATAC peaks determined within each cell type; cCRE association with a nearby gene is inferred by co-accessibility with promoters of nearby genes. Using the Cicero software, 15 cCREs co-accessible with the promoter were found for ACE2, whereas 73 were found for TMPRSS2 [22]. Given the dramatic changes in SARS-CoV2 infection, symptoms, and outcome risk across the age spectrum, the study quantified the expression and cCREs for ACE2 and TMPRSS2 across neonate, infant, and adult lungs. A larger proportion of AT2 cells expressed ACE2 and TMPRSS2 in adults compared to neonates and infants. Using the ATAC data, two cCRE clusters for TMPRSS2 in AT2 cells, comprising nine cCREs, showed enhanced accessibility with age. These clusters are associated with genes involved in response to viral infection, immune response, and injury repair and also overlapped with genes discovered in mouse models to associate with lung epithelial necrosis and chronic inflammation. Overall, this map of epigenomic/transcriptomic at SARS-CoV2 host genes can be a reference for studies using lungs from donors with SARS-CoV2 or animal models. A comprehensive transcriptomic/epigenomic atlas of cell-type resolved CREs (involved in transcriptional regulation) in the human lungs will facilitate the investigation of the gene regulatory mechanisms responsible for lung cell-type identity, function, and role in biological processes such as viral entry, as well as uncovering the effects of genetic variation on complex lung diseases.

A study utilizing single-cell RNA-Seq, single-cell ATAC-Seq (used to assess genome-wide chromatin accessibility), and spatial transcriptomics generated an atlas of human fetal lungs spanning 5–22 post-conception weeks (PCW) [23]. This study identified 144 cell types. Cell clusters are grouped by age groups into 5–6, 9–11, and 15–22 PCW. Many fetal cells were matched to their adult lung counterparts, such as fetal airway progenitors with adult secretory club cells and proximal secretory fetal cells with adult goblet cells. Fetal AT1 and AT2 cells had highly concordant transcriptomic profiles with adult cells. This study identified multiple cell types specific to the developing lung, such as progenitors of secretory cells and transition populations. Progenitor cells were further spatially localized. Epithelial progenitor cells were stratified into the tip, stalk, airway progenitor, and proximal secretory progenitor using spatial transcriptomics. They were spatially localized and assigned on a trajectory of differentiation. He et al. report that coupling single-cell methods with spatial analysis has allowed a comprehensive cellular survey of the epithelial, mesenchymal, endothelial, and erythrocyte/leukocyte compartments from 5–22 post-conception weeks [23]. The investigators identified previously uncharacterized cell states in all compartments, which include developmental-specific secretory progenitors and a subtype of neuroendocrine cell related to human small-cell lung cancer. The investigators also used this cell atlas to generate predictions about cell-cell signaling and transcription factor hierarchies which were tested in organoid lung models.

CellPhoneDB was utilized to elucidate cell-cell communication in distinct lung niches [24]. The airway niche comprised of fibroblasts, late airway SMCs, and airway epithelial cells. Cell-cell communication analysis between airway fibroblasts and airway epithelium included known signaling, such as via TGFB and BMP4, but also novel ones, such as FGF7/18 to FGFR2/3, and non-canonical WNT5A to FZD/ROR. These results were validated via tissue staining but also by using distal tip-based lung organoids; when grown in media with FGF, those organoids showed robust airway differentiation into secretory, basal, and ciliated cells. Using scATAC-Seq, transcription regulation was assessed in each cell type; analysis recapitulated known results, such as TCF21 in fibroblasts, KLF in secretory and AT1/AT2 cells, and TP63 in basal cells. A novel observation was that TCF4 was enriched in pulmonary neuroendocrine cells. Overall, this study illustrates how omics profiling of the lung, including single cell and spatial transcriptomics and ATAC-Seq, can provide rich references for diseases models used for integration with and interpretation of data generated from human, in-vitro, or in-vivo models of lung disease.

In addition to single-cell multiomics, omics related to cellular organelles, namely mitochondria and lysosomes, have been published that report multiomics in mitochondrial gene deletions and neural endolysosomal health [25,26,27].

## 3. Integration of Multiomics Data

While substantial knowledge can be derived from applying single omics in human samples or model organisms, a holistic and refined insight can be obtained via *multi-omics data.* Due to the large amounts of data from multi-omics, many integration workflows using machine learning have been reported, which reported on disease classification, subtyping, biomarker discovery and precision medicine [28,29,30,31,32]. Most projects in multi-omics data integration have stemmed from oncology; other fields, including cardiovascular medicine, follow suit [18,19,20,33,34,35].

Multi-omics data may be integrated via early, intermediate and late integration approaches (Figure 1) [36,37]. *Early integration* or early concatenation, although not complicated, may have problems with vast number of features while the number of available data points is low, known as the “curse of dimensionality” [7]. Multi-omics datasets may contain >50,000 features when the genome, transcriptome, and proteome are combined, but the available number of patient samples may be relatively small (hundreds or less). The heterogeneity of omics datasets may be a serious issue as omic data sets can have different distributions (e.g., numerical, categorical, continuous, discrete) and differ significantly in size (number of features). A necessary step in multi-omics analysis frequently is dimensionality reduction, which is reducing the number of variables to decrease the dimensionality and noise of a dataset. It is an optional simplification step, but some (early and intermediate integration) often require prior dimensionality reduction to be more effective. A summary of feature selection and feature extraction methods that decrease dimensionality has been published [28]. *The intermediate integration* strategy transforms each omics dataset independently into a simpler representation, thus overcoming some issues with the early integration strategy. Transformation converts the data set to a less dimensional and less noisy one, which decreases heterogeneity and facilitates integration and analysis. *Late integration* involves a combination of the results from each omics layer or each omics dataset by machine learning tools (or manually) and the predictions combined at the end [30,36]. Since each omics dataset is analyzed by omic-specific machine learning tools, the problems of noise and heterogeneity found in other strategies are not present. However, the downside of the late integration strategy is that it cannot capture inter-omics interactions and the different machine learning models (for the different omics datasets) do not share knowledge or utilize the complementarity information between omics [30]. Combining predictions is simply not enough to accurately exploit multi-omics data and understand the underlying biological mechanisms of diseases.

The study by Hoadley et al. is an example of machine learning integrative analyses where multi-omic integration of genomics, epigenomics, transcriptomics and proteomics in 12 different cancer types from human specimens in 6 different omics platforms, using a cluster of cluster arrangements (clusters from one omics platform were the input for the second level cluster analysis) helped to distinguish cancer subtypes [31]. New clinical outcome predictions in Chronic Lymphocytic Leukemia were possible using machine learning integration of somatic mutations, RNA transcription, and DNA methylation data [38]. Directly concatenating biomedical data with multi-omics with feature selection and machine learning can facilitate biomarker discovery [39]. The high complexity of the datasets calls for special computational approaches. Heterogeneity among datasets is a major problem when multi-omic datasets are integrated with large biomedical datasets. Many machine learning algorithms, including tree-based learning, multiple kernel-based learning and network bases approaches, have been reported to counter heterogeneity [40,41].

Another issue that needs to be addressed is missing data, and many imputational approaches have been proposed [28,42,43]. The reasons for missing data in multi-omics include low coverage of next-generation sequencing, low sensitivity in protein and peptide detection, and faltered metabolite measurement by tandem mass spectrometry [28]. Given that the biological samples are the same, it is statistically plausible to infer missing values in one omics from observed values and other omics by exploiting any existing correlations found through complete cases. The simplest approach to deal with missing data is a complete case analysis, also known as listwise deletion. Listwise deletion means that the entire sample is excluded from analysis if data is missing on any variable for that sample. However, it may result in substantial information loss if the missing data percentage is high. In addition to complete case analysis, traditional single imputation methods are very popular due to their ease of implementation. Any approach which estimates or guesses the missing values is called imputation. Missing values on a variable can be imputed by replacing it with a mean or median of the variable over all the available samples. Imputation based on regression or conditional mean imputation trains any regression model for the variable with missing data based on observed values. Subsequently, the model generates predicted values for the cases with missing data. The k-nearest neighbors approach is also commonly employed for imputing missing values. In general, it is recommended that any deterministic imputation should be done multiple times to account for the uncertainty in imputed values. Conventional single imputation methods for handling missing data include replacement with mean or mode values, hot-deck imputation, regression imputation, k-nearest neighbor, etc.

Maximum Likelihood approaches, including those based on an expectation-minimization (EM) algorithm and Direct Maximization, have attractive statistical properties compared to the conventional methods that often result in biased parameter estimates. Multiple imputation (MI) methods like Markov-chain Monte Carlo (MCMC) and multivariate imputation by chained equation (MICE) are also statistically robust, compared to conventional single imputation methods, as they take into account the uncertainty in the imputed values. MI for multiple factor analysis (MI-MFA) tackles the missing data problem in multi-omics analysis by performing MI based on hot-deck imputation. MI for nonlinear analysis can be performed using random forest (RF) and extreme learning machine (ELM). Adaptively-thresholded low-rank approximation (ALRA), singular value decomposition (SVD)-impute and SparRec methods employ matrix factorization for data imputation. In addition, imputation methods based on autoencoder and deep learning like denoising autoencoder-based MI (MIDA), AutoImpute and multilayer autoencoder (AE) have been proposed for high-dimensional datasets with missing data. Recently, integrative imputation methods such as ensemble regression imputation, multi-omics factor analysis (MOFA) and Late Fusion Incomplete Multi-View Clustering (LF-IMVC) are also available [28].

The issue of ‘class imbalance’ where rare events are the focus of our prediction, e.g., aggressive cancer or a severe rare event, needs to be tackled in our computational approaches. Methods used to address class imbalance includes data sampling, cost-sensitive learning and ensemble methods, which have been reviewed [28]. Class imbalance learning (CIL) methods are broadly classified into data sampling, cost-sensitive learning and ensemble methods [28]. Data sampling approaches balance the class distribution by either undersampling the majority class (e.g., random under sampling (RUS)), oversampling the minority class (e.g., synthetic minority oversampling technique (SMOTE)), or a combination of both (hybrid). Algorithm modification methods modify the learning algorithm generally by cost-sensitive weighting (e.g., Mnet, unbalance-aware network integration and prediction of protein functions (UNIPred), Spotlite and support vector machine (SVM)_weight). Cost-sensitive learning assigns a higher misclassification cost to minority class samples compared to majority class samples. Ensemble learning approaches like an ensemble with weighted majority voting, EasyEnsemble, Balanced Cascade, and ensemble weighted extreme learning machine (WELM) train multiple classifiers and aggregate their results to get the final output. Many existing integrative methods tackle imbalance by tuning models based on imbalance-aware evaluation measures. For example, data integration analysis for biomarker discovery using latent components (Diablo), super-layered neural network architecture (SNN), fuzzy pattern random forest (FPRF), and weighted majority voting (WMV) employ one or more CIL-specific evaluation measures like F-score, balanced error rate (BER), geometric mean (Gmean), Matthews correlation coefficient (MCC), area under precision-recall curve (auPRC), etc., instead of classification accuracy, to account for the bias introduced by imbalance in the dataset [28].

The potential applications are endless; we will enumerate a subset reported in the literature on lung-related disease models. Based on lessons from The Cancer Genome Atlas, molecular disease endotypes can be inferred for lung diseases. Disease drivers, disease presence or response to treatment biomarkers can be refined using multi-omics. Further, new therapeutic vulnerabilities can be determined and exploited by drug repurposing. Finally, multi-omics can be extended to surrogate sites, such as blood, skin, gut, saliva, or nasal cavity. In addition to the availability of numerous technologies, researchers have access to a trove of public data (reference or disease model datasets), including data in repositories such as LungMap, ENCODE, NIH Epigenome Roadmap, or NCBI Gene Expression Omnibus (GEO), or the Clinical Proteomic Tumor Analysis Consortium (CPTAC) [44,45,46,47]. The main challenges when integrating publicly available datasets from data repositories include data extraction, data integrity and scalability issues. Other issues relate to data structure differences across databases and data quality. The main challenges and limitations of multi-omic analysis studies are outlined in Table 2.

## 4. Lung Multiomics Models

In almost all respiratory diseases, the epithelium, a monolayer of cells which comprises the lining of the conducting and respiratory airways, is damaged. This compromises the proximal airways’ ability to warm, humidify, and cleanse the inhaled air and distal air to facilitate gas exchange. As a result, health and quality of life are severely impacted by the impaired lung function that occurs in respiratory disease. Human models, such as primary cells and organoids, and animal studies involving integrated multi-omics will allow differences in markers and biological processes between disease and non-disease models to be elucidated (Figure 2). These differences will provide insight into lung disease, including pathways that result in regeneration and repair. Understanding these pathways will be critical in developing preventive treatments and therapeutic modalities to treat lung diseases. It can eventually be harnessed to develop a personalized approach to treating respiratory diseases.

In recent decades, primary cells and transformed or tumor *cell lines* have been used to investigate lung diseases. The cells in these models retain many donor tissue characteristics and recapitulate markers and functions present in vivo [49,50,51]. These models have the advantage that they are amenable to genetic engineering, allowing the dissection of the role of individual molecules and pathways in disease [52]. Additionally, the ease of genetic engineering in these systems has allowed testing function via inducible gene expression [53]. Because of their wide use, many of these cell lines are well characterized, providing a foundation for multi-omics studies. Studies in cell lines are well suited for high throughput drug screening and evaluation of drug response [54] and are particularly valuable in studying lung cancer [55,56]. However, these models are not without limitations.

First and foremost, they fall short of replicating the complex nature of many respiratory diseases. Many lack the multiple cell types and cellular polarity present in the proximal and distal respiratory epithelium and exhibit an absence of morphology and structural features that play a significant role in lung biology. These cell lines also lack an immune cell component, which plays an important role in the etiology of many lung diseases [57]. Coupled with questions about the relevance of findings using these models, technical issues, including a requirement for tissue donors, a finite lifespan, and limited expansion capacity, have contributed to a reduced focus on using multi-omics in these models to study many respiratory diseases.

More recently, *organoid models* have come to the forefront of multi-omic studies of respiratory disorders. Organoids can be derived from either induced pluripotent stem cells or embryonic stem cells (hereafter referred to as iPSC organoids) or established from tissue-derived multipotent stem cells (referred to as ASC organoids) [58]. Differentiation of iPSC organoids occurs in a multistep process that involves a definitive endoderm stage, anterior foregut stage, and then into NKX2-1+ lung epithelial progenitors [59,60]. ASC organoids are established following mechanical and enzymatic isolation of conducting airway or respiratory epithelium stem cells from lavage, small amounts of native tissue, or biopsy specimen [61,62,63,64]. Both iPSC and ASC organoids rely heavily on manipulating exogenously added growth factors to induce differentiation of the mature polarized airway epithelium and the presence of extracellular matrix such as Matrigel^®^, synthetic matrices, or decellularized tissue scaffoldings. Both organoid models require cultivation on transwells under air-liquid interface conditions (ALI) where the basal side of the epithelium is in contact with media and the apical side is exposed to air to achieve maximum differentiation potential [63,65]. iPSC and ASC organoids can give rise to alveolar organoid models that recapitulate respiratory epithelium, nasal, trachea, or bronchial organoids that recapitulate conducting airway epithelium, and lung organoids that are a mixture [66,67,68]. Like primary and transformed cell models, organoids are amenable to genetic engineering and can be established from donors with genetic disorders that cause lung disease [69]. Organoids are well suited for drug screening and as models for infectious disease research [70,71,72,73] and recapitulate many aspects of other chronic lung diseases such as idiopathic pulmonary fibrosis [74] and cancer [63]. However, iPSC-derived airway cells do not seem to achieve the maturation levels observed in the human lung [75], although ASC organoids seem to contain mature epithelial cells, they lack stromal components such as the immune system that play a significant role in most lung diseases. However, the increased cellular complexity and the modeling of human epithelium combined with a forward-thinking multi-omics approach provide an area for advancement in information surrounding respiratory illness with significant translational potential.

Reports of machine learning integrated multiomics in organoid models and animal models have been published. One organoid model investigated the tumorigenic potential of alveolar type 2 cells (cells of origin of lung adenocarcinoma) using a multiomics and unsupervised machine learning approach [76]. The findings highlight the utility of understanding chromatin regulation in the early oncogenic versions of epithelial cells, which may reveal more effective means to intervene in the progression of Kras-driven lung cancer. In a murine model, authors investigated the interplay of the gut microbiome, metabolism, and host inflammation in obesity-associated asthma using a multiomics approach (microbiome, metabolomics and proteomics) to profile the gut-lung axis in the setting of allergic airway disease and diet-induced obesity [77]. The authors reported that changes to structural proteins in the lung airways and parenchyma may contribute to heightened lung elastance and serve as a potential therapeutic target for obese allergic asthma.

There are many animal models of lung diseases, including chronic diseases such as cystic fibrosis [78], idiopathic pulmonary fibrosis [79,80], viral and bacterial infections [81], and cancer [82]. Animal models of respiratory diseases offer several advantages including reproducibility, control of environmental factors, unlimited numbers of replicates, genetic phenotyping, and accessibility to lung tissue. Multi-omics approaches can be easily used to provide insight into the relationship between environmental stressors and the effect of the stressor on respiratory disease. The information gained can lead to detailed physiologic and pathologic pathways contributing to disease pathogenesis. Animal models of lung disease are instrumental in assessing the predisposition of genetic mutations in causing a specific disease and provide a model in which interactions between components of the whole system can also be examined [83]. Animal models are limited because, in most cases, there are significant differences between human lung tissue and animal lung tissue [44,84]. In addition, many human respiratory diseases are not recapitulated in animal models, and clinical manifestations are difficult to assess. However, comparisons between human and animal multi-omics analyses can validate animal models. Together, multi-omic-based approaches combining data collected from human in vitro and animal in vivo models will provide robustness, rigor, and reproducibility to support drawn conclusions.

## 5. Multiomics Insight into Clinical Disease

The major aim of multi-omic studies is to understand the disease pathophysiology of human disease. It is critical to extend in vitro studies, organoid models and animal models to studies in humans. These should be done longitudinally over time in the most pertinent age groups to understand better disease pathophysiology and biomarkers that can be utilized in the early detection and prevention of disease. Accurate and clinically relevant biological interpretation of multiomic analysis is paramount in understanding human disease. This is especially important in interpreting microbiome studies in human lung health and disease, as microbiota are normal colonizers of the respiratory tract [85].

### 5.1. Cystic Fibrosis

Cystic fibrosis is caused by a mutation in the chloride transporter gene, leading to thickened airway secretions [86,87]. The hallmark of the disease is chronic obstructive airway disease with infection and chronic inflammation [86,88]. However, persistent and heightened inflammation may be seen without infection [89,90]. Different types of clinical specimens have been evaluated in CF, which include upper airway secretions, sputum and bronchoalveolar fluid (BAL) [91,92,93] and exhaled breath [94,95]. Metabolites have been identified in BAL associated with inflammation and disease pathogenesis in CF [96,97,98]. O’Connor et al. compared CF patients’ BAL with control subjects to draw comparisons by network analysis [93].

Microbiome studies have pointed to complex polymicrobial communities in the airways and the lungs of patients with cystic fibrosis [91,92,93,99,100,101,102]. Although microbiome evaluations using 16S ribosomal RNA (rRNA) sequencing have pointed out bacterial communities in the airways (upper and lower and shifts with time) [100,103,104,105,106], it does not provide information on the community functionalities or metabolism. Whole genome sequencing of the microbiome enhances our understanding of the metabolic potential of the bacterial communities [92]. The availability of high throughput multi-omics technology will help us understand the microbe-microbe and microbe-host interplay that may determine disease severity and progression [4]. Identifying and profiling the metabolites, both host and microbial, by metabolomics will provide a view of the metabolic landscape and may identify biomarkers for disease diagnosis and prognosis [107].

A holistic approach integrating multi-omics that include microbiome WGS, metabolome, proteome and epigenome may provide deeper insights into the pathogenesis of the disease and aid in the management and improve clinical outcomes in CF patients [92,108]. Quinn et al. used molecular networking-based metabolomics to investigate the chemistry of CF sputa [92]. Investigators assessed whether the microbial metabolites detected reflected the microbiome and microbial cultures. Metabolites detected included xenobiotics, *P. aeruginosa* specialized metabolites and host sphingolipids. The clinical culture and microbiome profiles did not correspond to the detection of *P. aeruginosa* metabolites in the same samples. However, other investigators using multiomics investigations of sputa [91,92,101,102,109] have demonstrated correlations between the presence of pathogenic bacteria, metabolites and inflammation [99,109]. Twomey et al. identified strong correlations identified between the presence of strict anaerobes in sputum and the abundance of putrescine, pyruvate, and lactate [91]. Shi et al. proposed sparse multiple canonical correlation network analysis (SmCCNet) for integrating multiple omics data types along with a phenotype of interest and for constructing multi-omics networks (miRNA-mRNA networks that are specific to the phenotype [108]. The investigators demonstrated by simulations and published studies that SmCCNet had better overall prediction performance than popular gene expression network construction and integration approaches.

### 5.2. Chronic Obstructive Pulmonary Disease (COPD)

COPD is one of the leading causes of death in the United States. The Global Initiative for Chronic Obstructive Lung Disease (GOLD), a project initiated by the National Heart, Lung, and Blood Institute (NHLBI) and the World Health Organization (WHO), defines COPD as a “heterogeneous lung condition characterized by chronic respiratory symptoms (dyspnea, cough, expectoration, exacerbations) due to abnormalities of the airway (bronchitis, bronchiolitis) and/or alveoli (emphysema) that cause persistent, often progressive, airflow obstruction” [110] (www.goldcopd.org(accessed on 18 August 2023)). Substantial overlap exists between COPD and other disorders that may cause airflow limitation (e.g., emphysema, chronic bronchitis, asthma, bronchiectasis, bronchiolitis).

Yan et al. report airway microbe-host interactions in a study of 99 patients with COPD compared to 36 controls from China [111] for 2 endotypes, neutrophilic or eosinophilic inflammation. In neutrophil-dominant COPD, altered tryptophan metabolism leads to decreased indole-3-acetic acid (IAA), which affects host interleukin-22 signaling and epithelial cell apoptosis pathways. Yan et al. observed that airway microbiome-derived IAA mitigates neutrophilic inflammation, apoptosis, emphysema, and lung function decline via macrophage-epithelial cell cross-talk mediated by interleukin-22. Also, intranasal inoculation of two airway Lactobacilli restored IAA and recapitulated its protective effects in mice [111]. Multi-omic analysis in COPD also can help us understand the pathogenesis of pulmonary hypertension in COPD as it relates to heterogeneity and response to therapy [112], which may lead us to precision medicine- individualizing therapy and prognosis. Wang et al., using a multi-omic meta-analysis approach, used public COPD datasets (1640 16S evaluations and 26 samples from metagenomic sequencing) [113]. The investigators identified microbial shifts and established a global classifier for COPD using 12 microbial genera. Metabolic potentials of the airway microbiome were inferred and linked to host targets. 29.6% of differentially expressed human pathways were predicted to be targeted by microbiome metabolism [113].

COPD is a risk factor for lung cancer, and Sandri et al. report a multi-omic analysis of the lung stroma in patients with COPD who developed cancer compared to those who did not, testing the hypothesis that lung stroma in COPD has upregulated molecular mechanisms that support carcinogenesis [114]. Predictive variables for cancer, compared to the adjacent stroma, were mainly represented in the transcriptomic data, whereas predictive variables were associated with adjacent tissue compared to controls. Pathway analysis revealed extracellular matrix and phosphatidylinositol-4,5-bisphosphate 3-kinase-protein kinase B signaling pathways as essential signals in the tumor adjacent stroma [114]. A multi-omics approach to decipher complex pathways and networks of potential biomarkers will improve diagnosis, assist staging, decrease complications (PAH and cancer risk) and improve outcomes in COPD [115].

### 5.3. SARS-CoV-2 Infection

The Coronavirus Disease 2019 (COVID-19) is caused by Severe Acute Respiratory Syndrome Coronavirus 2 (SARS-CoV-2), and a pandemic has pervaded our work, productivity, and socioeconomic development. Xu et al., in a multi-omics analysis, showed that epithelial cells activated strong innate immune responses, including interferon and inflammatory responses [116]. Ubiquitinomics showed SARS-CoV-2 proteins were ubiquitinated during infection, although SARS-CoV-2 itself didn’t code any E3 ligase, and that ubiquitination at three sites on the Spike protein could significantly enhance viral infection. Therefore SARS-CoV-2 not only modulates innate immunity but also promotes viral infection by hijacking ubiquitination-specific processes, highlighting potential antiviral and anti-inflammation targets [116]. Unterman et al., in a single-cell multiomics analysis, of patients with COVID 19 assessed the immunology (T and B cell responses) and their response to tocilizumab, report desynchrony of the innate and adaptive immune interactions in progressive COVID-19 [117]. In a review, Li et al. summarize the multiomics integration-based molecular characterizations of COVID-19, which to date include the integration of transcriptomics, proteomics, genomics, lipidomics, immunomics and metabolomics to explore virus targets and developing suitable therapeutic solutions through systems biology tools (Figure 3) [118].

Wu et al. suing multi-omic analysis, report putative genes for covid severity (COVID-19 HGI using complementary CMO and S-PrediXcan methods), namely XCR1, CCR2, SACM1L, OAS3, NSF, WNT3, NAPSA, and IFNAR2 at five genomic loci [119]. Cantwell et al., in a hamster model infected with human SARS-CoV2, report the dynamic changes in gene transcription and protein expression over the course of the infection in a multi-organ kinetic analysis [120]. A multiomics study performed transcriptomic, epigenomic, and proteomic analyses that reported dysfunction in innate immunity in severe and fatal COVID-19 infection, including hyperactivation signatures in neutrophils and NK cells [121].

### 5.4. Lung Cancer and Lung Metastases

Lung cancer covers a broad spectrum; this review will focus on non-small cell lung cancer, including Lung Adenocarcinoma (LUAD) and Lung Squamous Carcinoma (LUSC). *Disease endotypes* in squamous cell lung cancers can be determined using transcriptomics and then characterized with several omics. The TCGA LUSC study identified four endotypes, classical, primitive, basal, and secretory, with DNA hypermethylation associated with the classical expression endotype [122]. The TCGA LUAD project identified three endotypes for lung adenocarcinoma: the proximal proliferative, with high rates of KRAS mutations. Proximal inflammatory, with co-mutations of NF1 and TP53, and the terminal respiratory unit, characterized by EGFR mutations and better prognosis [123].

*microRNA/mRNA networks* identify a subset of transcriptomic response that can be attributed to one mechanism of epigenomic dysregulation. microRNA targets can be inferred using several prediction engines, including TargetScan, mirDB, and DIANA-TarBase; miRNA/mRNA networks [124,125,126] can be inferred using algorithms such as SigTerms. miRNA/mRNA [124] were identified in LUAD [127] including hub miRNAs such as miR-539-5p, miR-656-3p, miR-2110, let-7b-5p, and miR-92b-3p; interestingly, other studies found further associations between LUAD, LUSC, miRNAs such as hsa-miR-195, hsa-miR-26b, and hsa-miR-126, and exercise [128]. An analysis of PanSCC cancers identified over-expressed miRNAs miR-205-5p and miR-944 across multiple squamous cancers, including LUSC, and further determined an association of their targets with epithelial-mesenchymal transition (EMT) [129]. A study using RNA-Seq transcriptomics and Mass-Spectrometry metabolomics in LUAD derived a 28-gene signature with prognosis capability. Also, they led to the identification of a novel lung cancer drug, AZD-6482, a PI3Kβ inhibitor [130].

Ho et al., in combined mass cytometry, immunohistochemistry, and RNA sequencing, identified the tumor microenvironment (TME) of lung metastases of pancreatic ductal adenocarcinomas (PDAC) [131]. The investigators report that the lung TME exhibits higher levels of immune infiltration, immune activation, and pro-immune signaling pathways, whereas multiple immune-suppressive pathways are emphasized in the liver TME. Sun et al. report the integration of extensive multiomics data sources, utilizing a total of 40 genome-wide functional annotations to prioritize and characterize single nucleotide polymorphisms (SNPs) that increase the risk of squamous cell lung cancer through the inflammatory and immune responses [132], including reanalysis of the ILCCO data. Their work highlights SNPs of genes associated with nuclear factor-κB signaling pathway genes and major histocompatibility complex-mediated variation in immune responses [132]. A multi-omics investigation involving whole exome sequencing (WES), RNA sequencing, methylation microarray, and immunohistochemistry (IHC) on eight pairs of non-small cell lung cancer (NSCLC) primary tumors and matched distant metastases [133] suggests that metastasis is a molecularly late event. Immunosuppression driven by different molecular events, including somatic copy number aberration, may be a common characteristic of tumors with metastatic plasticity [133].

### 5.5. Bronchopulmonary Dysplasia in Preterm Infants

Bronchopulmonary dysplasia (BPD) is a lung disease in preterm infants defined by oxygen requirements at 36 weeks postmenstrual age. Jensen et al. reported that the best BPD definition out of 18 prespecified evaluated definitions to predict death or serious respiratory morbidity through 18 to 26 months of corrected age was based on the mode of respiratory support administered at 36 weeks PMA, regardless of whether supplemental oxygen was used [134]. The conclusions were based on an evaluation of a prospective study from the National Institute of Child Health and Development (NICHD) network of 2677 preterm infants (GA < 32 weeks, 90 percent of the cohort were extremely preterm (EPT) with GA < 27 weeks) born between 2011 and 2015. The etiopathogenesis of BPD is multi-factorial, including disordered alveolar and microvascular development and inflammation leading to airway injury, inflammation, and parenchymal fibrosis. Multiomics may provide a holistic view of the molecular changes in BPD and may provide clues towards prevention or amelioration of the disease. The development of a molecular atlas of the developing lung (LungMAP) has been funded by the National Heart, Lung, and Blood Institute (NHLBI) [44,135]. This proposal will endeavor to create a molecular atlas of the developing lung (LungMAP), which will aid research and public education. The research will use multi-omics and other resources to study lung development, including interactive gene networks and dynamic cross-talk among multiple cell types to control and coordinate lineage specification, cell proliferation, differentiation, migration, morphogenesis, and injury repair. The information will benefit preterm infants with appropriate interventions to improve clinical outcomes, including survival and neurodevelopment.

Lal et al., in a study evaluating the functional metagenome of tracheal aspirates in preterm infants from 16srDNA data and metabolomics, reported differences in preterm infants who develop BPD [136]. The airway metabolome was enriched for metabolites involved in fatty acid activation and androgen and estrogen biosynthesis in BPD infants [136]. The role of exosomal microRNAs (miRNA) in BPD has been investigated and found that BPD-susceptible infants had reduced miR-876-3p in their tracheal aspirates [137]. A gain of function of miR-876-3p restores lung architecture in an animal model of BPD. The addition of lipopolysaccharide (LPS) in animal models leads to a decrease in miR-876-3p [137]. These results, in combination with the finding of increased abundance of Proteobacteria in tracheal aspirates of BPD infants, outlines the importance of lung dysbiosis and inflammation in etiopathogenesis [138,139]. Hyperoxia exposure of newborn mice serves as a model for BPD development in premature-born human babies; integration of RNA-Seq transcriptomics with miRNA targets revealed miR-30a as a potential driver for the reported sex disparity between males and females with respect to BPD risk [140,141]. In a mouse model of BPD, a study of lung microbiome and metabolome in 1–14 days old mice in response to exposure to hyperoxia and lipopolysaccharide (LPS) revealed that hyperoxia increased Intestinimonas abundance, whereas LPS decreased Clostridiales, Dorea, and Intestinimonas; further integration with a published lung transcriptomics signature of hyperoxia derived a gene signature with biomarkers potential for risk of BPD development [142]. The importance and relevance of lung T-cell multi-omic interactions with the genome, epigenome and microbiome have been reviewed in the context of BPD in preterm infants (Figure 4) [143].

### 5.6. Pulmonary Hypertension

Pulmonary arterial hypertension (PAH) is a heterogeneous and highly morbid disease defined as mean pulmonary artery pressure (mPAP) of >20 mm Hg obtained by cardiac catheterization [144]. The recent change in definition stems from a significant increase in mortality and hospitalization risk with mPAP >20 mm Hg. Pulmonary hypertension may result from left heart disease; lung diseases and/or hypoxia; pulmonary artery obstructions (particularly thromboembolic syndromes); and undifferentiated or multifactorial causes, including sickle cell disease and sarcoidosis but PH attributable to left heart and lung diseases are the most prevalent subtypes [144]. Multiomics investigations into this highly heterogeneous disease will increase our understanding of the pathophysiology and derive biomarkers for monitoring and improving patient outcomes. Chen et al. studied the association between gut microbiota composition and host metabolome signatures in a left pulmonary artery ligation (LPAL)-induced PH rat model and reported significant gut dysbiosis in LPAL-PH rats in association with specific changes in gut and lung metabolome profiles [145]. In another multiomic study, Konigsberg 2021 et al. studied molecular signatures in an idiopathic pulmonary fibrosis model [146], and Titz et al. investigated the multiomics of toxic effects of aerosols in mouse models [147]. Hong et al. performed co-expression analysis by RNA sequencing 96 disease and 52 control samples from the lung biobank [148]. The investigators report a co-expression module of 266 genes associated with pulmonary arterial hypertension (PAH) severity, such as increased PVR and intimal thickness, but also with compensated PAH, such as lower hospitalizations, WHO functional class and NT-proBNP. Modules of co-expressed genes were identified by ‘Weighted Gene Co-Expression Network Analysis (WGCNA) from the biobank lung RNA-seq samples. The lung transcriptome was grouped into clusters based on gene co-expression, referred to as modules, representing co-regulation or shared biological functions. The authors investigated whether a pattern matching the pink module signature with known pharmacologic and genetic perturbation signatures could reveal novel therapeutic targets. The pink module signature was compared against 8559 perturbation signatures from Connectivity Map (CMap) [149]. These CMap signatures were grouped into 171 classes that share similar mechanisms of action or biological functions, thereby identifying pharmacological targets. A pharmaco-transcriptomic screen discovered ubiquitin-specific peptidases (USPs) as potential therapeutic targets [148].

## 6. Societal and Ethical Issues Related to the Use of Multiomics and Machine Learning in Healthcare [150]

The intent of using multi-omics and machine learning is to improve diagnosis or predict clinical phenotypes requiring intervention, which will translate to better patient outcomes. However, we should also be mindful of the potential adverse consequences of the use of AI and ML in healthcare as they are increasingly explored for patient care. Societal issues related to fairness, explainability, privacy, ethics and legislation should be sufficiently addressed in new projects [151]. Fairness is a central perspective in healthcare, and the models developed by AI/ML should be fair to human characteristics, including gender, race and ethnicity. For example, a predictive model developed with one gender, race, or ethnic group only is not likely to apply to the population at large and is bound to fail in the real world. Privacy is another critical issue in healthcare. Patients’ anonymity and privacy should be respected if the ML modelling is performed using private information of the patients, e.g., identifiers such as medical record numbers, social security numbers or even the zip code where they live. Ethical research is the cornerstone of medicine and healthcare, and several questions need to be addressed. Will it be possible for AI and ML models to follow or address the basic biomedical ethical principles of respect for autonomy, non-maleficence, beneficence and justice? Could the use of AI, if the model is poorly developed, cause more harm than good? Could AI, in some way their predictions, increase psychological stress in parents and make the patients poor insurance clients if we predict long-term poor outcomes early? Are there ethical concerns, conflicts of interest, or individuals on the part of the researcher or clinician, hospital or institution—financial or otherwise? [152]. How are medico-legal risks that arise from the use of AI managed? [153] We, as proponents of the use of AI/ML, should make sure that we address societal and ethical issues adequately to maximize risk-benefit ratios.

## 7. Summary

The availability and accessibility of high-throughput multi-omics technologies, including microbiome evaluation of polymicrobial communities in the airway and the lung, transcriptomics, metabolomics, proteomics and genome-wide evaluation approaches, have increased our understanding of the systems biology approach. These technologies contribute to a better holistic understanding of the etiopathogenesis of many respiratory diseases. Multiomic strategies should provide insights into predictive analytics. Along with early detection of phenotypic changes, multiomic approaches that provide biomarkers that are present before the development of disease will provide opportunities for early intervention, e.g., early detection of inflammatory conditions (inflammatory bowel disease) *and* allergic and autoimmune diseases. Single-cell and organelle multi-omic approaches are being increasingly used, including single-cell imaging and spatial omics. Integrating molecular with non-molecular data through machine learning, clinical (e.g., electronic medical records), imaging, phenotypic and socioeconomic correlates (exposures) is essential to complete our holistic understanding of human health and disease. Integrating such diverse data calls for novel and appropriate computational approaches requiring a multi-disciplinary collaborative approach among molecular scientists, clinicians, bioinformaticians and computational scientists. Ultimately, this knowledge will open avenues for novel preventative and therapeutic strategies to treat airway diseases and contribute to novel and innovative research areas that continue to improve human health and well-being.

## Figures and Tables

**Figure 1 microorganisms-11-02116-f001:**
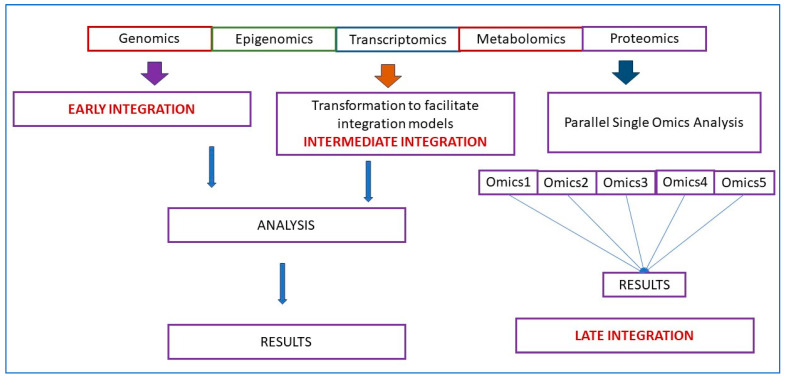
Multi-omics integration.

**Figure 2 microorganisms-11-02116-f002:**
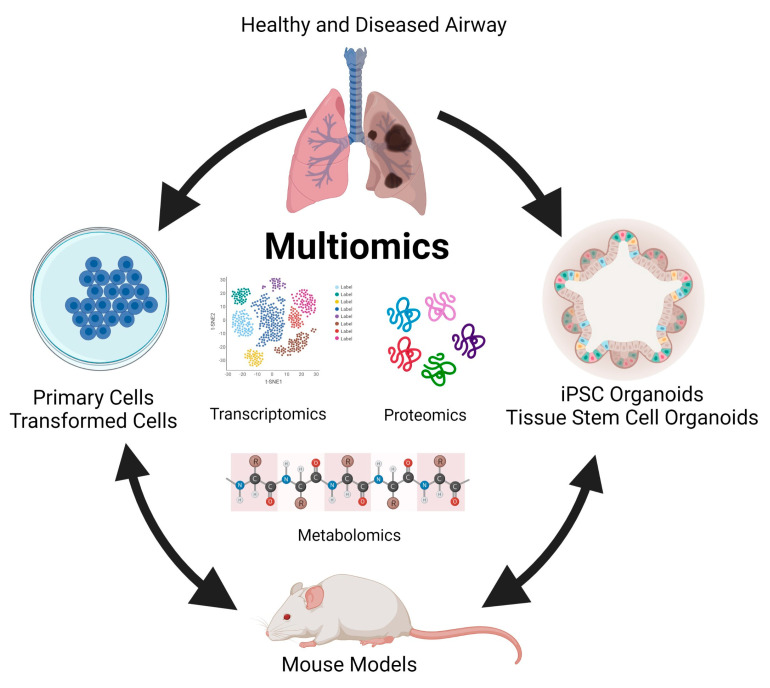
Modelling multi-omics studies in the study of human lung disease.

**Figure 3 microorganisms-11-02116-f003:**
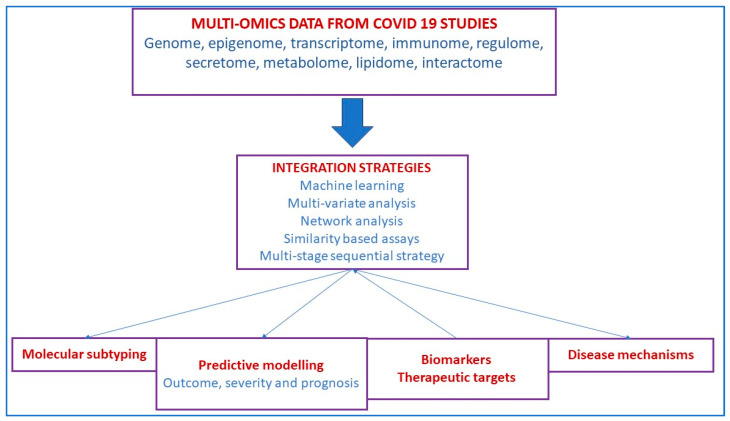
Characterization of COVID-19 infection using multi-omics.

**Figure 4 microorganisms-11-02116-f004:**
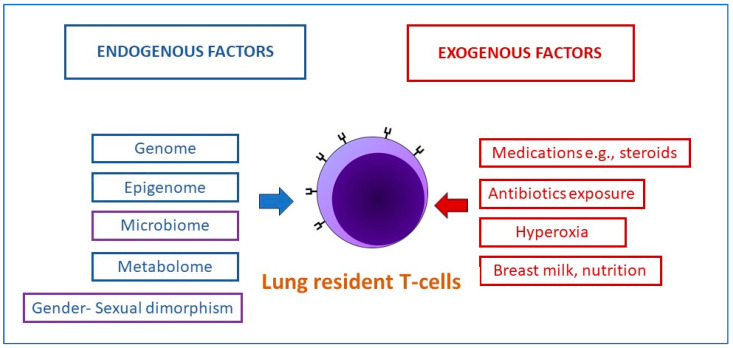
Multi-omic T-cell interactions in the lung in bronchopulmonary dysplasia.

**Table 1 microorganisms-11-02116-t001:** Omics Definitions and Descriptions.

“Omic” Technology	Description
Genome	**Genomics** focuses on identifying genetic variations associated with disease, response to treatment, or prognosis. Genome-wide associations (GWAS) have successfully explained complex phenotypes in human diseases (GWAS catalogue https://www.ebi.ac.uk/gwas/home (accessed 18 August 2023)).
Epigenome	**Epigenomics** focuses on the genome-wide characterization of reversible modifications of DNA or DNA-associated proteins, such as DNA methylation or histone acetylation, which are major regulators of gene transcription and cellular fate. Those modifications can be influenced both by genetic and environmental factors, can be long-lasting, and are sometimes heritable.
Transcriptome	**Transcriptomics** focuses on genome-wide mRNA transcription qualitatively (which transcripts are present, identification of novel splice sites, RNA editing sites) and quantitatively (how much of each transcript is expressed). A small amount of RNA is transcribed for protein synthesis, and a much larger amount is encoded for other purposes, which may be implicated in disease.
Proteome	**Proteomics** quantifies peptide abundance, modification, and interaction. Specific peptides may be helpful in diagnosis, monitoring or prognostication of disease and may function as disease biomarkers. Mass spectroscopy has revolutionized the field of proteomics not only for quantifying peptides but also for identifying functionality mediated by post-translational modifications, including proteolysis, glycosylation, phosphorylation, nitrosylation, and ubiquitination.
Metabolome	**Metabolomics** quantifies multiple small molecules, including amino acids, fatty acids, carbohydrates, or other products of cellular metabolic functions. Metabolite levels and relative ratios reflect metabolic function, and out-of-normal range perturbations often indicate disease.
Microbiome	**Microbiomics** focuses on the abundance and composition of microbioal communities in humans and their association with health and disease. Human skin, mucosal surfaces, and the gut are colonized by microorganisms, including bacteria, viruses, and fungi, collectively known as the microbiota (and their genes constituting the microbiome).

**Table 2 microorganisms-11-02116-t002:** Challenges in multi-omics analysis (adapted from Tarazona 2021 [48]).

Challenges in Multi-Omics Analysis	Possible Solutions
Data collection issues Non-uniform missing data	Proper methods of missing value imputation Missing value-compatible imputation methods
2. Integration analysis issues Inefficient computation Diverse signal/noise Poor biological interpretation	Effective migration to the cloud and GPU systems and community training Analysis methods to balance power and experimental design Better visualization tools and statistical models
3. Dissemination or sharing of data with the community Distributed data storage Inconsistent sample annotation	Recommendations on multiomics experimental standards and standard data annotation and structure

## Data Availability

No new data were created or analyzed in this study. Data sharing is not applicable to this article.

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
