# Peer review of "Multiomic Investigations into Lung Health and Disease"

_microorganisms, 2023, doi:10.3390/microorganisms11082116_

Round 1

Reviewer 1 Report (New Reviewer)

The current manuscript by Blutt et al reviews and discuss the use of multi-omic technologies in the biology of lung diseases, where data integration with these tools could be useful for diagnosis and prognosis. Overall, the manuscript shows a valuable perspective, but some concerns were found: 

Major comments

1.     The hypothesis and/or the main objective of the review are not listed neither in the abstract nor in the hypothesis sections. Please do.

2.     Authors need to clearly define which studies were completed in primary cells, organoids, animal models or human samples. For example, lines 163-170 three studies are discussed but none of them are clearly explained. Was the data described in Hoadley et al obtained from patients? What is a cluster of clusters? How was the data integrated? What is the importance of distinguish cancer subtypes?

3.     What do the authors mean in the last 2 paragraph of cystic fibrosis section? In the current version of the manuscript there is a combination between general samples used in omics and specific study descriptions, please provide more detail of the studies mentioned.

4.     The COPD section is built as a list of studies with its respective conclusion. The section lacks on a disease description and more importantly the data is not critically discussed. Finally, please include a conclusion.

5.     Similarly, define pulmonary hypertension, including physiological pathogenesis and please discuss how multi-omic technologies could improve the study or diagnosis of this disease.

6.     The manuscript would benefit if a bacterial pneumonia section is included. Mult-omic technologies have been extensively used to study host-pathogen interaction in mouse models but also in human samples.

Minor comments

1.     Please avoid include other reviews as reference.  Include original research manuscripts.

2.     Define abbreviations and contractions.

3.     Please use the same format in titles and subtitles.

4.     Please add references statements from lines 36-39 and 39-42. 

5.     Please use the same reference format along the manuscript (see lines 87 and 100).

6.     English quality must be improved (i.e. line 163-166, line 186-190, line 192-194).

7.     Figure 2 is confusing. What do the arrows mean? 

8.     Please use the same format when an author is mentioned, see lines 385-386 and 445.

English quality must be improved (i.e. line 163-166, line 186-190, line 192-194).

Author Response

Reviewer 2 Report (New Reviewer)

The authors have made an interesting attempt at “Multiomic Investigations into Lung Health and Disease.” The manuscript is interesting; however, the authors need to justify the scientific writing manuscript. Some of the general comments are provided below:

1.     What specific respiratory diseases are included in the 5 million deaths reported yearly?

2.     How do the recent advances in high-throughput technologies help provide a holistic view of pathophysiology in lung diseases?

3.     How is multiomics data integrated with clinical and social data to improve health outcomes?

4.     Can you provide specific examples of how multiomic integrations have led to testable hypotheses about mechanisms in lung diseases?

5.     What are the main findings of the single-cell multi-omics study on SARS-CoV2 infection, and how do they relate to age-associated risks and outcomes?

6.     How do epigenomic/transcriptomic maps of SARS-CoV2 host genes serve as a reference for further studies involving lungs from donors with SARS-CoV2 or animal models?

7.     What insights were gained from the atlas of human fetal lungs generated using single-cell RNA-Seq, single-cell ATAC-Seq, and spatial transcriptomics?

8.     How can missing data be addressed in multi-omics datasets, and what are the implications of class imbalance in predictive modeling for rare events?

9.     Can you provide specific examples of studies that have successfully applied machine learning integration in multi-omics data analysis, particularly in lung-related disease models?

10. What are the main challenges or limitations faced when integrating public datasets from repositories like LungMap, ENCODE, NIH Epigenome Roadmap, or NCBI Gene Expression Omnibus (GEO)?

11. Are there any ethical considerations when using multi-omics data, particularly in the context of personalized medicine and patient privacy?

12. Can you provide more details about Hong et al.'s co-expression analysis by RNA sequencing in the lung biobank? How were the 266 genes associated with pulmonary arterial hypertension (PAH) severity and compensated PAH identified, and what potential therapeutic targets were discovered?

13. Were there any limitations or challenges faced in these multiomic studies, and how were they addressed?

Author Response

Reviewer 3 Report (New Reviewer)

The manuscript is a generalised overview of omics usage in the domain of lung health. It does not have a more specific focus but relies more strongly on molecular methods. The manuscript reports on various methods, but somehow falls short of the most recent and rather promising advances. I would suggest that the authors look into more detail regarding the organelles and provide maybe a wider perspective on this. The second part that is very missing is the need to improve the primary studies in humans. This must be the central message regarding the difficulties in defining the phenotypes of interests in a harmonized way. It is amazing once you get to read seeing how vastly different various authors grasp some concepts and measure things differently. Maybe you can suggest the need for harmonization across the field in order to talk about the same things.  Sadly, the focus on microorganisms and microbiome is only a peripheral, so maybe that oart can be expanded to a bit more detail. Maybe you can mention the microbiome difficulties in the analysis after discovering pathogenic bacteria in health subjects? This would indeed provide grounds for claims that we need more primary studies and a better understanding of the entire field before we can make substantial progress. Also, I am missing some progressive ideas – usually after reading a manuscript of this kind, the reader should have a clear idea of the field, and be able to delve into new areas and concepts. This is somehow missing, maybe you can include a few more futuristic studies and suggest how those may/will impact the field of research? 

Round 2

Reviewer 1 Report (New Reviewer)

I appreciated all the amendments made by the authors. 

Most of my comments were addressed although authors should pay attention to minor format/typos issues through the manuscript (miilion, iintermediate, etc) as well as citations (Konigsberg 2021 et al in hypertension section). 

In my opinion, these issues should be addressed in the proof-correction step.

I would like to point out that Dr(c) Pedro Silva ([email protected]) assisted me in the review process of this manuscript

Reviewer 2 Report (New Reviewer)

The authors have addressed my queries and now the manuscript is acceptable for publication. 

This manuscript is a resubmission of an earlier submission. The following is a list of the peer review reports and author responses from that submission.

Round 1

Reviewer 1 Report

The topic of the paper very interesting. I thoroughly enjoyed reading the review. It  is well-written, in my opinion. very useful to find all the strategies grouped together.

Author Response

Many thanks for your review and feedback. No specific points to address.

Reviewer 2 Report

Blutt et al., in their review, provide an overview of the various lung diseases that have utilized omics technologies to address mechanisms underlying signaling deficits, identify biomarkers and therapeutic targets. While this review mainly cites and touches upon studies that have used omics, it lacks a thorough understanding of how integration of omics is achieved. This is particularly since omics integration is a major topic of the paper.

But this review fails to provide relevant information on how integrations of omics is achieved to tease out biomarkers, specific signaling pathways and actionable targets. Integration of two or more omics datasets is NOT trivial as is generally assumed. The authors should devote a proper section that highlights with examples how integration of omics data was done. E.g. one great example of explaining integration of omics data is that which includes RNAseq, Proteomics and PTMomics (phosphoproteomics) datasets. RNA and protein levels do not exhibit linear correlation yet, studies have reported both. Furthermore, proteomics (global protein-level expression) and phosphoproteomics for instance cannot be directly correlated. How can the authors provide suitable information that can help explain the integration of such datasets.

Other points:

The authors are strongly advised to carefully proofread their manuscript as grammatical errors persist throughout the manuscript. e.g. “Single omics is limited by the only providing…”

Also, “…and iintermediate integration”

Please carefully proofread manuscript again.

Author Response

The authors should devote a proper section that highlights with examples how integration of omics data was done. 

We expanded the section on multiomics integration as suggested by the reviewer. Our aim was to provide a broad overview but understand multi-omics integration is an important section.

Other points: addressed in the manuscript

Reviewer 3 Report

The presented draft aims to review the possibilities of lung pathology/ physiology monitoring using -omics techniques. Since the XXI gave us lots of sophisticated techniques to track almost every single pathway of the metabolism, the usage of -omics techniques is key to developing new therapies and broadening knowledge. 

I went through the paper very carefully, with a very friendly approach, albeit, still found major issues that preclude publishing the paper in the present form. Please follow my majors:

1) The whole paper is written in a careless manner that is not suitable for scientific writing. There are copy-pasted links in the manuscript body, and there are unnecessary bold fonts, italic fonts, and underlies. There is no page numbering so I can not point out the exact examples. Even there are double affiliations for number 4 and there is no affiliation for number 5. Part of the text looks like it was copy-pasted from Notepad or other writing software.  

2) Even for narrative reviews, there is a very limited amount of information given regarding the following PRISMA guidelines and description of the searching strategy, which is so crucial for review papers. The Author should include at least: Data sources and searches, Study eligibility criteria, Study selection process, Data extraction, and study quality assessment (assessing the risk of bias (ROB) for each included study), Data synthesis. MeSH terms (in addition/replacement of keywords) are necessary to be included. For each step, it is necessary to explain to the reader with pictures or tables. It is necessary to explain what was drawn at each step to lead to the result. Moreover, a figure showing the PRISMA-based workflow must be drawn accordingly to the Prisma schema. After that, a discussion is valuable even for narrative papers. A description of the Data Mining strategy should also be included. 

3) The paper is very chaotic - there is no logical workflow, suggesting that the presented draft is a mosaic structure with a pretty hard-to-digest form. I am truly surprised that the Authors discuss only a few (not even major!) lung diseases and forgot about so basic conditions like influenza virus infection and many others. 

4) Presented figures do not bring any information for the Readers. These graphs must be redone to make them scientifically sound.

5) Data included in the table is misleading. Omics technique name is i.e. Metabolomics; Genomics. The one written by the Authors is not correct. 

6) There is no clinical trial data included. Please follow the currently processed data from clinicaltrials.gov

7) There are no limitations to the paper discussed.

Dear colleagues, I must say with regret, that I feel that the presented draft was written too fast without any plan and further meaning. I would be more than happy to see a fixed and resubmitted manuscript.  

The used English is correct, albeit, it sounds like a mosaic structure of styles and subtypes of English. It is very hard to digest it. 

Author Response

The presented draft aims to review the possibilities of lung pathology/ physiology monitoring using -omics techniques. Since the XXI gave us lots of sophisticated techniques to track almost every single pathway of the metabolism, the usage of -omics techniques is key to developing new therapies and broadening knowledge. I went through the paper very carefully, with a very friendly approach, albeit, still found major issues that preclude publishing the paper in the present form.

Please follow my majors:

1) The whole paper is written in a careless manner that is not suitable for scientific writing. There are copy-pasted links in the manuscript body, and there are unnecessary bold fonts, italic fonts, and underlies. There is no page numbering so I cannot point out the exact examples. Even there are double affiliations for number 4 and there is no affiliation for number 5. Part of the text looks like it was copy-pasted from Notepad or other writing software. 

We are sorry that this reviewers feels this way. The bolding and the underlines are to emphasize some points. Page numbering was not (?) a format for micro-organisms. We have since removed all the unnecessary bolding, italics and underlining. Not sure about page numbering. We do not agree with his comments on 'copy pasted' from notepad OR chaotic flow (please see reviewer 1 comments on the ease of readability).

2) Even for narrative reviews, there is a very limited amount of information given regarding the following PRISMA guidelines and description of the searching strategy, which is so crucial for review papers. The Author should include at least: Data sources and searches, Study eligibility criteria, Study selection process, Data extraction, and study quality assessment (assessing the risk of bias (ROB) for each included study), Data synthesis. MeSH terms (in addition/replacement of keywords) are necessary to be included. For each step, it is necessary to explain to the reader with pictures or tables. It is necessary to explain what was drawn at each step to lead to the result. Moreover, a figure showing the PRISMA-based workflow must be drawn accordingly to the Prisma schema. After that, a discussion is valuable even for narrative papers. A description of the Data Mining strategy should also be included. 

This is a broad overview on a hot topic and the reviewer's comments on PRISMA guidelines or search strategy is not valid. The senior author has published >15 systematic reviews of which some were Cochrane reviews and feels that use of PRISMA guidelines is not indicated for this overview.

3) The paper is very chaotic - there is no logical workflow, suggesting that the presented draft is a mosaic structure with a pretty hard-to-digest form. I am truly surprised that the Authors discuss only a few (not even major!) lung diseases and forgot about basic conditions like influenza virus infection and many others. 

We are sorry that this reviewers feels this way (please see reviewer 1 comments on the ease of readability). The overview was not meant to be comprehensive, so we included one infectious etiology (SARS CoV1) and other common conditions including a neonatal condition.

4) Presented figures do not bring any information for the Readers. These graphs must be redone to make them scientifically sound.

We do not agree with this reviewer on the figures. We think it portrays the broad overview of the subject.

5) Data included in the table is misleading. Omics technique name is i.e. Metabolomics; Genomics. The one written by the Authors is not correct. 

We do not agree with this reviewer on the table comment. We think it describes the different components what multiomics are.

6) There is no clinical trial data included. Please follow the currently processed data from clinicaltrials.gov

 clinical trial.gov  is a place for ongoing trials and trial data is usually not available unless the published report is included. In this broad review, clinical trials’ data where current ongoing trials are listed may not be the focus of this paper.

7) There are no limitations to the paper discussed.

We added a sentence on limitations on this review not being comprehensive since it is a wide area of coverage.

Comments on the Quality of English Language

The used English is correct, albeit, it sounds like a mosaic structure of styles and subtypes of English. It is very hard to digest it. 

The author group is an experienced group that has published more than 100 scientific papers in reputed journals. We have carefully revised the manuscript to have a single style.

Round 2

Reviewer 3 Report

The Authors did not address correctly most of the major comments nor included requested changes. 

Their explanation for not following PRISMA guidelines is against the proper way of publishing unbiased, well-sourced review papers. 

There are still unnecessary bold/italic/underlied fonts used.

I truly appreciate that the Authors published over 100 scientific papers, but it doesn't mean that every paper is ideal. The process of peer review is to increase the quality, integrity, and readability of the draft/manuscript. Here, the process doesn't work, thus, I am not able to recommend any further processing of the manuscript. 

With kind regards.